# Context-aware Heterogeneous Graph-driven Multimodal Representation Learning for Emotion Recognition

## Abstract

Multimodal emotion recognition (MER) aims to infer human affect from verbal, vocal, and visual signals, a core challenge in representation learning for human–AI interaction. State-of-the-art approaches, including standard Transformers and graph-based models, often collapse modalities into uniform structures, ignoring modality-specific temporal dynamics and asymmetric dependencies. We propose a novel context-aware heterogeneous graph-driven representation learning that explicitly encodes both structural and semantic heterogeneity. Each modality is first contextualized with dedicated Transformer encoders, enriching unimodal features before graph construction. We then introduce a relation-aware graph transformer that performs type-conditioned message passing, enabling specialized transformations across sequential, cross-modal, and speaker-conditioned edges. The topology is adapted to the target regime: in multi-party dialogue (IEMOCAP, MELD), we distinguish within-speaker and cross-speaker temporal flows, while in single-speaker videos (CMU-MOSEI), we extend k-step temporal links to capture offset dynamics. In both settings, co-temporal edges synchronize audio, visual, and textual cues. Experiments demonstrate consistent gains over prior state-of-the-art, showing that structural and semantic heterogeneity are indispensable for robust multimodal representation learning. Our results establish that explicitly modeling interaction structure, rather than relying on generic sequence attention, is critical for advancing multimodal learning. To support reproducibility and further research, we will release our source code.

## 1 Introduction

Emotion is inherently multimodal and is conveyed simultaneously through spoken language, vocal prosody, body pose, and facial expression. Systems that aspire to respond empathetically, such as social robots, conversational agents, or digital mental health tools, must therefore ground their decisions in *multimodal representations*. The key challenge is how to fuse signals that are *temporally asynchronous*, *semantically asymmetric*, and *structurally heterogeneous*. Early pipelines adopted *early fusion* by concatenating modality-specific features or encoding them with a shared backbone to jointly learn embeddings. However, they often entangle heterogeneous data and blur modality-specific dynamics Tzirakis et al. (2017). In contrast, *late-fusion* trains modality-specific classifiers, for example, fastText for text, 1D CNNs for audio, 2D CNNs for vision, and merges their logits with a meta-learner, preserving autonomy but discarding fine-grained interactions Dixit & Satapathy (2024). To bridge this gap, intermediate architectures are introduced with richer but still shallow couplings. The compact bilinear pooling captures multiplicative correlations between video and audio embeddings with reduced memory cost Nguyen et al. (2018), while hierarchical kernels combine EEG and peripheral physiology at multiple spatial scales Zhang et al. (2021). However, CNN-based models remain limited in modeling long-range dependencies and asynchronous emotion cues.

Transformers have effectively become the default backbone for multimodal fusion. Cross-modal attention aligns speech and vision streams to predict arousal valence Huang et al. (2020), while standard encoders such as RoBERTa, Wav2Vec, and FAb-Net combined with multimodal transformers achieve state-of-the-art (SotA) across benchmarks Siriwardhana et al. (2020). Extensions that include a learnable waveform Dutta & Ganapathy (2022), adaptive attention between the BERT and

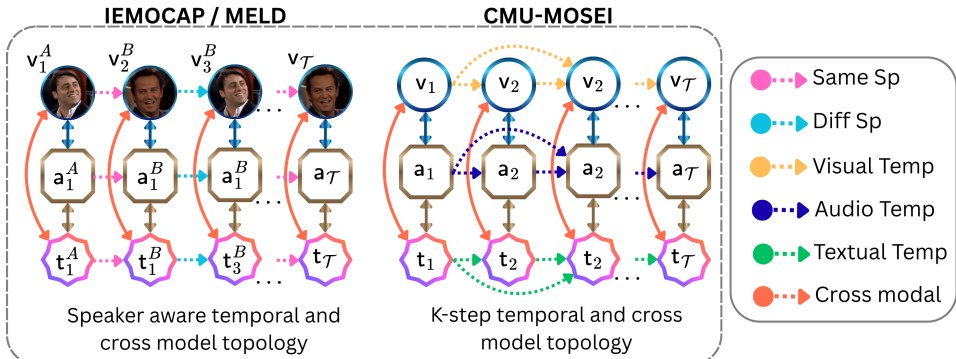

**Figure 1: Left (IEMOCAP/MELD):** dialogue topology with per-utterance tri-modal nodes—visual $v$, audio $a$, text $t$—coupled vertically (cross-modal) and linked temporally within each modality (orange/blue/green). Speaker links connect same-speaker turns (pink) and cross-speaker turns (cyan). **Right (CMU-MOSEI):** single-speaker sequence with the same $k$-step intra-modal temporal edges and per-step cross-modal coupling (here $k$=2), but no speaker links.

CNN branches Makhmudov et al. (2024), and cross-attention guided by static facial cues Zhang et al. (2022) further demonstrate the versatility of attention-based fusion. However, *vanilla transformers attend indiscriminately*, often merging misaligned signals and overlooking structured dependencies.

Graph neural networks (GNNs) offer a principled alternative by explicitly encoding interactions: observations as nodes, and temporal adjacency, speaker turns, or semantic alignment as edges. Homogeneous GNNs such as DialogueGCN Ghosal et al. (2019), GraphMFT Li et al. (2023b), MMGCN Hu et al. (2021b), GraphCFC Li et al. (2023a), and DEDNet Wang et al. (2024) outperform transformer-based fusion in conversational MER. However, these models route all messages through a single weight matrix, ignoring *relation semantics* and *modality identity*. Recent heterogeneous graph methods alleviate this bottleneck. For example, HMG-Emo personalizes affect recognition by weaving user and image nodes Bhattacharyya et al. (2024), while MMPGCN Meng et al. (2024) and HHGN Peng et al. (2025) add speaker-aware or hierarchical relations. Similarly, ES-IHGNN integrates event–state interactions for richer conversational cues Zha et al. (2024). Other approaches extend heterogeneity to physiological signals Jia et al. (2021); Wang et al. (2025); Liu et al. (2024). Despite progress, these models are often tailored for sequence-level tasks, handle temporal and cross-modal links in isolation, or propagate messages with *uniform attention weights*.

**Motivation and contribution:** This landscape reveals a clear gap; existing fusion methods fail to *jointly capture structural heterogeneity* (temporal vs. cross-modal vs. speaker) and *semantic heterogeneity* (relation type, modality identity) within a *unified representation framework*. Addressing this gap is essential for advancing multimodal representation learning for downstream tasks.

In this work, we present a context-aware heterogeneous graph framework for multimodal emotion recognition that unifies structural and semantic heterogeneity within a single representation learning paradigm. Our key contributions are: (1) *Fine-grained heterogeneous graph construction:* We represent each utterance as a graph whose edges are typed by relation, sequential speaker-aware links, k-step temporal offsets, and co-temporal cross-modal alignments. This enables explicit modeling of temporal, cross-modal, and speaker-conditioned dynamics. (2) *Coupling modality-specific encoders with relation-aware message passing:* Dedicated transformer encoders to enrich unimodal signals before graph construction. A relation-aware graph transformer then performs edge-type-conditioned message passing, learning specialized transformations for each interaction type, rather than applying uniform aggregation. (3) *Unified design across regimes:* We adapt graph topology to both multi-party dialogue (IEMOCAP, MELD) and single-speaker video (CMU-MOSEI), showing that the same framework generalizes across distinct multimodal settings. (4) *SotA performance with structural interpretability:* Experiments with widely used MER benchmarks demonstrate consistent and significant improvements over previous methods. Beyond accuracy gains, our framework highlights the importance of explicitly modeling relation types, providing interpretable insights into how emotions emerge from multimodal interactions.

## 2 RELATED WORK

**Transformers and Cross-Modal Attention:**   Transformer architectures dominate multimodal fusion by offering token-level alignment and long-range dependencies. HyFusER Yi et al. (2025) applies bidirectional attention between the KoELECTRA text and the HuBERT speech. It significantly improves Korean emotion recognition while limiting itself to two modalities and requiring an ensemble prediction. TACFN Liu et al. (2025) prunes redundant tokens with self-attention and merges streams through adaptive blocks, reducing parameters by 30% in RAVDESS, leaving speaker dynamics unmodeled. A Capsule-Graph-Transformer combines ViT faces, CapsNet text, and GCN audio into hierarchical networks Filali et al. (2025), but rigid modality-specific branches restrict flexibility. More recent work, such as TDTN-HLFR, disentangles modality-common vs. modality-specific cues and reconstructs missing streams with SimSiam-based twins, but its dual-stage training raises complexity Li et al. (2025). Despite these advances, vanilla self-attention attends indiscriminately and often combines temporally misaligned cues, motivating structure-aware alternatives.

**Homogeneous Graph Neural Networks:**   Graph neural networks (GNNs) introduce relational reasoning, but most rely on homogeneous topologies. MMGCN builds fixed modality-wise graphs, capturing global context at the expense of redundant edges Hu et al. (2021b). MM-DFN employs dynamic modality gating Hu et al. (2022a), and DGSNet uses dual graphs to separate private from shared cues Tang et al. (2023), but both still aggregate messages without edge semantics. DEDNet highlights salient nodes through dual-level attention Wang et al. (2024), but its uniform edges prevent fine-grained modeling. Transformer-driven MultiEMO Shi & Huang (2023) and contrastive UniMSE Hu et al. (2022b) bypass explicit graphs for speed, sacrificing interpretability. These approaches highlight the promise of relational learning but fail to capture modality- and relation-specific interactions.

**Heterogeneous Graph Neural Networks:**   Heterogeneous GNNs explicitly distinguish node and edge types, allowing richer modeling. GraphCFC adds directed cross-modal edges, but struggles to scale beyond three modalities Li et al. (2023a). HMG-Emo links users and images for the social media affect Bhattacharyya et al. (2024), but conversational turn-taking remains ignored. MMPGCN introduces homogeneity-aware edge weighting, improving IEMOCAP/MELD accuracies while underfitting complex semantics Sun et al. (2020). ESIHGNN enriches graphs with emotional state nodes and COMET-based knowledge Zha et al. (2024), though pre-processing increases overhead. HHGN separates the directed context graph from the undirected cross-modal graphs Peng et al. (2025), achieving SotA accuracy in IEMOCAP, but doubling memory use and omitting the hierarchy of emotion classes. GraphSmile integrates sentiment dynamics for robustness against abrupt changes Li et al. (2024b). Extensions to physiological signals HetEmotionNet Jia et al. (2021), DHGRNN Wang et al. (2025), and VBH-GNN Liu et al. (2024) demonstrate the value of heterogeneity, although generalization to conversational MER remains limited. Despite clear gains, current heterogeneous GNNs typically treat temporal and cross-modal links in isolation or apply uniform attention, leaving structural–semantic heterogeneity underexploited.

**Robustness and Cognitive Augmentation:**   Beyond architectural fusion, recent efforts focus on robustness under missing modalities and cognitively inspired augmentation. Distillation-based approaches such as Decoupled Multimodal Distilling (DMD) separate homogeneous and heterogeneous spaces with dual-graph distillation, improving the alignment of CMU-MOSEI but requiring four auxiliary losses Li et al. (2023c). RMER-DT imputes missing modalities with conditional diffusion, excelling under 50% dropout, yet increasing inference latency Zhu et al. (2025). Contrastive frameworks such as UniMSE enforce modality invariance with supervised contrastive loss Hu et al. (2022b). However, a heavy data augmentation is required. These methods strengthen generalization, but largely sidestep explicit relational modeling. In addition to robustness, cognitively inspired methods incorporate external memory or episodic reasoning. COGMEN uses memory to track speaker states, improving contextual coherence but scaling poorly with dialogue length Joshi et al. (2022), while CFN-ESA fuses episodic and semantic attention for hierarchical cues, but lacks explicit graph reasoning for fine-grained interactions Li et al. (2024a). Together, these directions highlight the usefulness of auxiliary mechanisms but reveal a common limitation: they improve resilience or context modeling while sidestepping explicit relational reasoning across modalities.

## 3 PROPOSED METHOD

**Problem Formulation:**   We formulate multimodal emotion recognition as a structured prediction problem over dialogues. Given a dialogue $D = \{u_1, \ldots, u_{|D|}\}$ with $|D|$ utterances, each utterance

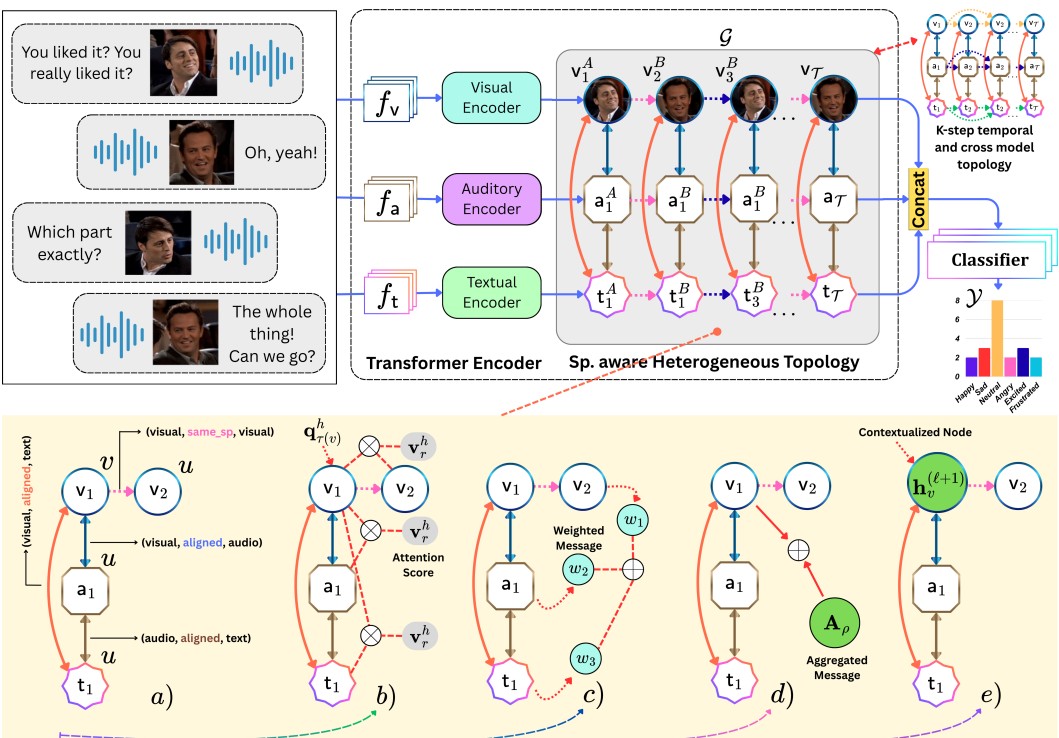

**Figure 2: Overview.** visual (v), acoustic (a), and textual (t) features are first extracted by modality-specific encoders. The $\mathcal{T}$ embeddings are then treated as nodes of a heterogeneous graph whose typed edges capture (i) speaker aware or k-step temporal links inside each modality and (ii) alignment links across modalities. A stack of heterogeneous graph layers refines these node states. After the final layer, the three modalities are concatenated and fed to a classifier that outputs the discrete emotion label $\mathcal{Y}$. **Bottom row (one graph layer).** **(a)** Example neighbourhood of a visual node showing the three edge types. **(b)** Relation-specific attention generates per-edge messages. **(c)** Messages are aggregated across heads and relations. **(d)** The aggregated message is added to the node's residual and normalised, yielding **(e)** the contextualised representation passed to the next layer.

$u_t$ is observed in three modalities: text (t), audio (a), and vision (v). For each modality $m \in \{t, a, v\}$, we pre-extract a feature matrix $f^m \in \mathbb{R}^{|D| \times d_m}$, where $d_m$ is the input feature dimension for modality $m$. Speaker identity (when available) is also provided for each utterance, enabling us to capture multi-speaker conversational dynamics. We learn a predictor $f_\theta(f^t, f^a, f^v) = y \in \mathcal{Y}$, where $\mathcal{Y}$ is the set of emotion categories. Unlike conventional fusion strategies, our approach explores heterogeneous, relation-aware graph representations to explicitly encode temporal, cross-modal, and speaker-specific interactions before classification.

**Dialogue–Level Heterogeneous Graph:** We represent each dialogue as a heterogeneous graph $\mathcal{G} = (\mathcal{V}, \mathcal{E})$, where every utterance corresponds to one node per modality, and each node is assigned a type $\tau(v) \in \{t, a, v\}$. This construction explicitly couples multimodal signals at the utterance level with typed relations that capture speaker dynamics, temporal flow, and cross-modal synchrony.

*Speaker-aware temporal edges.* For multi-speaker dialogues, we create directed edges between utterances conditioned on speaker identity: (i) *past_same* edges connect utterances of the same speaker (e.g., $u_1^A \to u_3^A$, where both utterances are spoken by speaker A), modeling inter-speaker consistency; (ii) *past_other* edges connect utterances across speakers (e.g., $u_2^A \to u_4^B$), capturing cross-speaker influence and conversational turn-taking. This allows the graph to represent the dialogue structure beyond the raw sequence order.

*Sliding temporal edges.* For monologue-style dialogues, where speaker turns are absent, we instead connect utterances within a bounded temporal horizon. With step size $k$; each utterance $u_t$ is connected to $u_{t+1}, \ldots, u_{t+k}$ and their reverse for bidirectional, producing an efficient temporal receptive field that preserves *local context* without quadratic scaling in sequence length.

*Cross-modal edges.* Across all dialogue types, we add bidirectional edges between nodes of different modalities in aligned utterance indices (e.g., $\mathsf{t} \leftrightarrow \mathsf{a}$, $\mathsf{a} \leftrightarrow \mathsf{v}$, $\mathsf{t} \leftrightarrow \mathsf{v}$). These enforce *synchronous alignment* across modalities, allowing the model to integrate complementary cues at the utterance level.

In summary, our graph topology adapts to both dyadic conversations (via speaker-aware temporal edges) and monologues (via sliding temporal edges), ensuring a unified framework that generalizes across multi-party dialogue (e.g., IEMOCAP, MELD) and long-form monologue datasets (e.g., CMU-MOSEI).

**Transformer Encoder:** Before graph-level reasoning, we enrich each modality sequence with a Transformer encoder to obtain *contextualized unimodal utterance representations*. For modality $m \in \{\mathsf{t}, \mathsf{a}, \mathsf{v}\}$, the utterance sequence $[f_1^m, \dots, f_{|D|}^m]$ is projected to a shared hidden dimension $d_m$, producing $\mathbf{x}_t \in \mathbb{R}^{d_m}$ for each utterance index $t$. Each Transformer layer applies multi-head self-attention. For head $h$, we compute: $\mathbf{q}_t^h = \mathbf{W}_Q^h \mathbf{x}_t$, $\mathbf{k}_j^h = \mathbf{W}_K^h \mathbf{x}_j$, $\mathbf{v}_j^h = \mathbf{W}_V^h \mathbf{x}_j$, where $\mathbf{W}_Q^h, \mathbf{W}_K^h, \mathbf{W}_V^h \in \mathbb{R}^{d_q \times d_m}$, $d_q = d_m/H$ and $H$ is the number of heads. Attention scores and weights are $\alpha_{t,j}^h = \frac{\langle \mathbf{q}_t^h, \mathbf{k}_j^h \rangle}{\sqrt{d_q}}$ and $\widetilde{\alpha}_{t,j}^h = \mathrm{softmax}_{j=1,\dots,|D|} \alpha_{t,j}^h$, resulting in an aggregated context $\mathbf{z}_t^h = \sum_{j=1}^{|D|} \widetilde{\alpha}_{t,j}^h \mathbf{v}_j^h$. The multi-head output is concatenated and projected: $\mathbf{z}_t = \mathbf{W}_O \big\|_{h=1}^{H} \mathbf{z}_t^h$. A residual connection with layer normalization (LN) and a feedforward network (FFN) complete the layer: $\mathbf{y}_t = \mathrm{LN}(\mathbf{x}_t + \mathbf{z}_t)$ and $\mathbf{h}_t = \mathrm{LN}(\mathbf{y}_t + \mathrm{FFN}(\mathbf{y}_t))$. Stacking multiple layers produces contextualized utterance embeddings $\mathbf{h}_t$, which serve as the initial node features for the heterogeneous graph. This ensures that *intra-modal temporal context* is captured before cross-modal and cross-speaker reasoning.

**Heterogeneous Graph Encoder:** Our dialogue graph $\mathcal{G} = (\mathcal{V}, \mathcal{E})$ is processed by a stack of heterogeneous graph layers. At layer $\ell$, each node $v$ is represented by $\mathbf{h}_v^{(\ell)} \in \mathbb{R}^{d_h}$ (output of Transformer encoder). Its neighbourhood $\mathcal{N}(v)$ is defined by the edge construction (Fig. 1): for multi-speaker dialogues, this includes past utterances from the same or different speakers; for monologues, sliding temporal neighbors $k$; in all cases, $\mathcal{N}(v)$ also contains aligned cross-modal counterparts.

*Relation-aware attention computation* (Fig. 2b): For each neighbor $u \in \mathcal{N}(v)$, relation $r = (\tau(u), \tau(v))$, and head $h$, we calculate relation-specific queries and keys: $\mathbf{q}_v^h = \mathbf{Q}_{\tau(v)}^h \mathbf{h}_v^{(\ell)}$ and $\mathbf{k}_{u \to v}^h = \mathbf{K}_r^h \mathbf{h}_u^{(\ell)}$, with projection matrices conditioned either on node type $\tau(v)$ or relation type $r$ and $\mathbf{Q}_{\tau(v)}^h, \mathbf{K}_r^h \in \mathbb{R}^{d_q \times d_h}$ ($d_q = d_h/H$). The attention weights are then $\alpha_{u \to v}^h = \langle \mathbf{q}_v^h, \mathbf{k}_{u \to v}^h \rangle / \sqrt{d_q}$, $\widetilde{\alpha}_{u \to v}^h = \mathrm{softmax}_{u \in \mathcal{N}(v)} \alpha_{u \to v}^h$, and the weighted message from the neighbor $u$ is $\mathbf{m}_{u \to v}^h = \widetilde{\alpha}_{u \to v}^h \mathbf{V}_r^h \mathbf{h}_u^{(\ell)}$, with $\mathbf{V}_r^h \in \mathbb{R}^{d_q \times d_h}$. Importantly, unlike homogeneous graph-attention (GAT) layers, our projection matrices are relation- or type-specific and shared between layers, allowing the encoder to learn *distinct transformations for temporal, cross-modal, and speaker-conditioned edges*.

*Relation-wise aggregation* (Fig. 2c–e): Messages are first concatenated across $H$ heads, then mapped via a relation-specific ($r$) transform $\mathbf{W}_r$. Summing over all relation types yields the updated node embedding: $\mathbf{h}_v^{(\ell+1)} = \mathrm{LN}\big(\mathbf{h}_v^{(\ell)} + \sigma(\sum_{r \in \mathcal{R}_v} \mathbf{W}_r \big\|_{h=1}^{H} \sum_{u \in \mathcal{N}_r(v)} \mathbf{m}_{u \to v}^h)\big)$, where $\sigma$ is RELU. Stacking $L$ layers propagates information across temporal edges, speaker turns, and cross-modal alignments.

**Classifier:** For each utterance, we concatenate the final embeddings across modalities: $\mathbf{z} = [\mathbf{h}_{\tau_1}^{(L)}; \dots; \mathbf{h}_{\tau_{|\mathcal{T}|}}^{(L)}] \in \mathbb{R}^{|\mathcal{T}|H}$. A shared two-layer MLP maps $\mathbf{z}$ to task-specific logits, with a dropout layer preceding the final projection. Training minimizes cross-entropy loss, and inference applies the SoftMax function over the logits. In the multitask setting, both prediction heads share the same representation $\mathbf{z}$.

## 4 EXPERIMENTAL RESULTS AND DISCUSSION

**Datasets:** We evaluate on three widely used multimodal emotion recognition benchmarks: **IEMO-CAP** Busso et al. (2008), **MELD** Poria et al. (2018), and **CMU-MOSEI** Zadeh et al. (2018). To ensure comparability with prior work, we adopt the official public splits released by MMGCN Hu et al. (2021b) and CFN-ESA Li et al. (2024a) for fair comparison. **IEMOCAP** provides dyadic

conversations with 5,810 training and 1,623 test utterances annotated with six emotions (happy, sad, neutral, angry, excited, frustrated). **MELD** extends to multi-party conversations with 11,098 training and 2,610 test utterances across seven emotions (neutral, surprise, fear, sadness, joy, disgust, anger). **CMU-MOSEI** contains long monologue-style videos with multimodal annotations for sentiment and emotion, offering a challenging large-scale benchmark. Utterances are represented by modality-specific features. Text: RoBERTa$_{base}$ embeddings (1,024-dim) fine-tuned with COSMIC Ghosal et al. (2020). Audio: openSMILE IS10 (1,582-dim) for IEMOCAP and log-Mel filterbanks (300-dim) for MELD. Vision: DenseNet-121 FER+ activations (342-dim). For MOSEI, we use the aligned multimodal features released by GraphSMILE Li et al. (2024b). We report weighted $F_1$ and accuracy as primary metrics, consistent with prior multimodal emotion recognition work Hu et al. (2022a); Majumder et al. (2019); Ghosal et al. (2019); Wang et al. (2024); Li et al. (2024b).

**Implementation Details:** Our model stacks $L=3$ heterogeneous graph layers with $H=4$ heads. The hidden sizes are 256 (IEMOCAP, MELD) and 640 (CMU-MOSEI), resulting in 14M, 15.1M, and 48M parameters, respectively. Training is implemented in PyTorch 2.0.0 + PyG 2.6.1 on a single NVIDIA A40 (48GB); peak memory usage remains under 0.8GB. Optimization uses AdamW with cosine annealing and warm restarts. The initial learning rate is $1 \times 10^{-5}$; batch size/epochs are 16/200 (IEMOCAP, MELD) and 16/50 (CMU-MOSEI).

## 4.1 Comparison with the State-of-the-Art Methods

**Results on Emotion Recognition:** We benchmark against strong transformer-style models (UniMSE Hu et al. (2022b), Joyful Zheng et al. (2023), SACL-LSTMHu et al. (2023), M3NetChen et al. (2023)), dialogue-specific systems (DialogueCRNHu et al. (2021a), MM-DFNHu et al. (2022a), COGMENJoshi et al. (2022)), and recent graph-based methods (DER-GCNAi et al. (2024), DCGCNYang et al. (2024), MMGCNHu et al. (2021b), CFN-ESALi et al. (2024a), GraphSmileLi et al. (2024b)). The results are reported in Tables 1–3.

*IEMOCAP*: Our model achieves **73.26 / 73.25** (**ACC / WF1**), surpassing the strongest prior heterogeneous-graph method (GraphSmile: 72.77 / 72.81) by **+0.49 / +0.44**. We also exceed the best transformer baseline (Joyful: 71.03 WF1) by **+2.22**, establishing a new state-of-the-art (SotA).

*MELD*: We obtain **64.71 / 65.03**, competitive but below GraphSmile (67.70 / 66.71). However, we outperform the transformers (Joyful: 62.53 / 61.77; SACL-LSTM: 64.52 / 64.55) and dialogue-specific models (MMGCN, MM-DFN) by sizeable margins (up to +6.72 WF1). This suggests that our framework generalizes across modalities but remains challenged by multi-party complexity.

*CMU-MOSEI*: Our model delivers **46.96 / 44.33 WF1** (**ACC / WF1**), matching or exceeding SotA: we obtain the best ACC among the listed methods (+0.14 over GraphSmile), with WF1 close to theirs (–0.6). We also improve substantially over transformer baselines (M3Net: +3.29 / +3.21) and over dialogue-specific systems such as MMGCN (+1.29 / +0.22) and MM-DFN (+1.67 / +1.35).

Across three benchmarks, our model achieves SotA or near-SotA performance, consistently outperforming transformer- and dialogue-specific baselines. The gains are most pronounced on IEMOCAP and MOSEI, demonstrating the effectiveness of explicitly modeling structural and semantic heterogeneity. While GraphSmile excels on MELD by explicitly modeling emotion shifts (intra- and inter-modal fusion), our method achieves stronger performance on IEMOCAP and MOSEI by leveraging relation-aware message passing that generalizes across dialogue and monologue regimes.

**Results on Sentiment Analysis:** We benchmark against transformer-style (SACL-LSTM, M3Net), dialogue-specific (DialogueCRN, MM-DFN), and graph-based (MMGCN, GraphSmile) systems on IEMOCAP, MELD, and CMU-MOSEI (Table 4).

*IEMOCAP*: Our model achieves **85.15 / 85.38** (**ACC / WF1**), surpassing GraphSmile (84.97 / 85.09) by +0.18 ACC / +0.29 WF1. We also outperform the best transformer (M3Net: +0.86 / +1.19) and dialogue-specific systems such as DialogueCRN (83.43 / 83.52) and MMGCN (82.05 / 82.94).

*MELD*: We obtain 73.71 / 73.69, competitive with the strongest baseline DialogueCRN (74.18 / 73.96, –0.47 / –0.27). However, we exceeded all graph-based baselines (GraphSmile: +0.11 / +0.29; MMGCN: +0.15 / +0.23) and transformers (M3Net: +0.68 / +0.37).

*CMU-MOSEI*: Our model delivers 66.84 / 65.92, setting a new best on both metrics. We slightly outperform GraphSmile (66.83 / 65.69) and MMGCN (66.66 / 65.89), while achieving large gains

**Table 1:** Main Results for the MERC Task on IEMOCAP Dataset

| Method | Happy | Sad | Neutral | Angry | Excited | Frustrated | ACC | WF1 |
|---|---|---|---|---|---|---|---|---|
| MetaDrop[†] | – | – | – | – | – | – | 69.38 | 69.59 |
| UniMSE[†] | – | – | – | – | – | – | 70.56 | 70.66 |
| COGMEN[†] | 51.90 | 81.70 | 68.60 | 66.00 | 75.30 | 58.20 | 68.20 | 67.60 |
| EmoLR[†] | – | – | – | – | – | – | 68.53 | 68.12 |
| Joyful[†] | 60.94 | **84.42** | 68.24 | 69.95 | 73.54 | 67.55 | 70.55 | 71.03 |
| DER-GCN[†] | 58.80 | 79.80 | 61.50 | **72.10** | 73.30 | 67.80 | 69.70 | 69.40 |
| DCGCN[†] | – | – | – | – | – | – | – | 68.31 |
| DialogueCRN[‡] | 53.85 | 82.66 | 71.03 | 62.33 | 77.64 | 58.81 | 68.70 | 68.82 |
| MMGCN[‡] | 47.10 | 81.91 | 66.44 | 63.51 | 76.17 | 59.06 | 67.10 | 66.81 |
| MM-DFN[‡] | 43.36 | 83.23 | 70.03 | 70.19 | 73.11 | 64.01 | 69.44 | 68.83 |
| SACL-LSTM[‡] | 51.30 | 82.25 | 71.39 | 67.78 | 75.26 | 66.94 | 70.55 | 70.60 |
| M3Net[‡] | 60.93 | 78.84 | 70.14 | 68.06 | 77.11 | 67.42 | 70.92 | 71.07 |
| CFN-ESA[†] | 53.67 | 80.60 | 71.65 | 70.32 | 74.82 | 68.06 | 71.04 | 70.78 |
| GraphSmile[†] | **63.09** | 83.16 | 71.07 | 71.38 | **79.66** | 66.84 | 72.77 | 72.81 |
| **Ours** | 61.15 | 81.24 | **73.62** | 69.64 | 78.84 | 69.54 | **73.26** | **73.25** |

The marker [†] denotes results reported in the original publications, whereas [‡] denotes results obtained from our replication experiments. Underlined values indicate the second-best and bold values indicate the best result.

**Table 2:** Main Results for the MERC Task on MELD

| Method | Neutral | Surprise | Fear | Sadness | Joy | Disgust | Anger | ACC | WF1 |
|---|---|---|---|---|---|---|---|---|---|
| AGHMN[‡] | 76.40 | 49.70 | 11.50 | 27.00 | 52.40 | 14.00 | 39.40 | 63.50 | 58.10 |
| Joyful[†] | 76.80 | 51.91 | – | 41.78 | 56.89 | – | 50.71 | 62.53 | 61.77 |
| HU-Dialogue[†] | – | – | – | – | – | – | – | 61.38 | 58.56 |
| DialogueCRN[‡] | 76.15 | 56.72 | 18.67 | 38.29 | 63.21 | 27.69 | 50.67 | 62.38 | 63.32 |
| MMGCN[‡] | 78.62 | 57.78 | 3.77 | 40.35 | 63.60 | 12.20 | **53.68** | 60.42 | 58.31 |
| MM-DFN[‡] | 79.84 | 58.43 | 15.79 | 31.65 | 64.01 | 28.04 | 53.60 | 62.49 | 59.46 |
| SACL-LSTM[‡] | 77.42 | 58.50 | 20.41 | 39.58 | 62.76 | **34.71** | 52.08 | 64.52 | 64.55 |
| GraphSmile[†] | **80.35** | **59.11** | 18.18 | **42.46** | **64.99** | 32.43 | 53.67 | **67.70** | **66.71** |
| **Ours** | 77.07 | 58.77 | **24.00** | 41.89 | 63.64 | 30.48 | 49.68 | 64.71 | 65.03 |

over transformer baselines (M3Net: +2.12 / +4.26; SACL-LSTM: +11.56 / +21.42) and dialogue-specific models (DialogueCRN: +12.73 / +17.42; MM-DFN: +0.49 / +1.32).

Our heterogeneous graph-driven framework achieves new SotA results on IEMOCAP and MOSEI, and remains competitive on MELD. It consistently improves over transformer- and dialogue-specific baselines, while slightly surpassing recent graph-based methods in sentiment analysis.

## 4.2 ABLATION STUDY

**Effect of Modalities:** We conduct a full factorial modality ablation on IEMOCAP, MELD, and CMU-MOSEI to quantify each modality's contribution (Table 5). Across all benchmarks, *tri-modal fusion* (V+A+T) consistently achieves the best performance for both emotion and sentiment, confirming strong cross-modal complementarity. For example, on IEMOCAP we obtain 73.26 WF1 / 73.25 ACC (emotion) and 85.15 / 85.38 (sentiment); on MELD, 65.03 / 64.71 and 73.69 / 73.71, and on CMU-MOSEI, 44.33 / 46.96 and 66.84 / 65.82.

Among unimodal inputs, text dominates, especially for sentiment tasks, but incorporating prosody and vision consistently improves performance. For example, on IEMOCAP sentiment, A+T improves to 84.90 / 84.81, and adding vision reaches the trimodal optimum. The bimodal results follow a stable ordering of A+T > V+T > V+A, emphasizing the complementary role of speech cues. Interestingly, the magnitude of the fusion gains varies by dataset: largest on IEMOCAP (dyadic, emotionally rich), moderate on MELD (multi-party, more overlapping speech), and small-est on CMU-MOSEI(monologue-style, language-dominant). This indicates that while *language is the strongest* standalone modality, *audio* and *vision* provide *dataset-dependent but complementary signals*, and the hierarchy ordering *trimodal > bimodal > unimodal* holds robustly across bench-marks.

**Effect of Feature Dimensionality:** We further study the impact of feature dimensionality on per-formance across IEMOCAP, MELD, and CMU-MOSEI (Table 9 Appendix). Results show that intermediate dimensions (256) consistently yield the best balance between expressiveness and gen-

**Table 3:** Main Results for the MERC Task on CMU-MOSEI

| Method | HN | N | WN | Neutral | WP | P | HP | ACC | WF1 |
|---|---|---|---|---|---|---|---|---|---|
| COGMEN[†] | – | – | – | – | – | – | – | 43.90 | – |
| DialogueCRN[‡] | 0.00 | 4.29 | 7.98 | 25.09 | 51.80 | 3.22 | 0.00 | 37.88 | 26.55 |
| MMGCN[‡] | 0.00 | 19.51 | _43.75_ | **42.45** | 54.64 | _36.13_ | 0.00 | 45.67 | 44.11 |
| MM-DFN[‡] | 0.00 | 16.98 | 37.94 | _39.64_ | 56.58 | 32.51 | **8.51** | 45.29 | 42.98 |
| SACL-LSTM[‡] | 0.00 | 0.00 | 0.00 | 17.87 | 55.28 | 0.00 | 0.00 | 38.60 | 25.95 |
| M3Net[‡] | 0.00 | 12.50 | 37.26 | 33.29 | 56.10 | 33.94 | 0.00 | 43.67 | 41.12 |
| GraphSmile[†] | 0.00 | **28.79** | 43.50 | 39.19 | _57.30_ | 35.95 | 6.25 | **46.82** | **44.93** |
| **Ours** | **06.90** | _20.85_ | **45.94** | 29.73 | **57.34** | **42.03** | **23.64** | **46.96** | **44.33** |

The marker [†] denotes results reported in the original publications, whereas [‡] denotes results obtained from our replication experiments. The abbreviations "HN" and "WN" denote Highly Negative and Weakly Negative, respectively, and other sentiment labels follow by analogy.

**Table 4:** Main Results for the MSAC Task on **IEMOCAP**, **MELD**, and **CMU-MOSEI**.

| Method | IEMOCAP | | | | | MELD | | | | | CMU-MOSEI | | | | |
|---|---|---|---|---|---|---|---|---|---|---|---|---|---|---|---|
| | Negative | Neutral | Positive | ACC | WF1 | Negative | Neutral | Positive | ACC | WF1 | Negative | Neutral | Positive | ACC | WF1 |
| DialogueCRN[‡] | 87.87 | 67.09 | _89.96_ | 83.43 | 83.52 | 69.34 | **80.05** | 66.67 | **74.18** | **73.96** | 20.87 | 12.71 | 70.51 | 54.11 | 48.50 |
| MMGCN[‡] | 87.07 | 68.17 | 88.31 | 82.75 | 82.94 | 69.36 | 79.03 | 66.60 | 73.56 | 73.46 | **58.10** | 38.62 | 78.47 | 66.66 | _65.89_ |
| MM-DFN[‡] | **89.08** | 71.92 | 87.85 | 84.60 | 84.69 | _70.83_ | 77.89 | 66.05 | 73.30 | 73.27 | _53.67_ | 36.11 | _78.67_ | 66.35 | 64.62 |
| SACL-LSTM[‡] | 88.52 | 72.95 | 89.85 | 84.85 | 85.05 | 69.16 | 79.08 | 66.67 | 73.56 | 73.44 | 10.59 | 0.00 | 71.73 | 55.28 | 44.50 |
| M3Net[‡] | 87.52 | 68.35 | **91.93** | 84.29 | 84.19 | 69.44 | 78.46 | _67.12_ | 73.33 | 73.32 | 53.08 | 25.92 | 77.64 | 64.72 | 61.66 |
| GraphSmile[†] | 88.28 | _72.89_ | 89.93 | _84.97_ | _85.09_ | 70.08 | _79.15_ | 64.84 | 73.60 | 73.40 | 56.14 | _39.80_ | 78.18 | _66.83_ | 65.69 |
| **Ours** | _88.69_ | **73.99** | 89.32 | **85.15** | **85.38** | 79.10 | 67.77 | **69.25** | _73.71_ | _73.69_ | 53.31 | **40.87** | **78.74** | **66.84** | **65.92** |

eralization. With 256, our model achieves peak or near-peak performance: 73.26 WF1 / 73.25 ACC (emotion), 85.15 / 85.38 (sentiment) on IEMOCAP, 65.03 / 64.71 and 73.69 / 73.71 on MELD, and 44.33 / 46.96 and 66.84 / 65.82 on CMU-MOSEI. Larger dimensions (512, 768) offer no further gains and sometimes degrade performance (e.g., MELD emotion at 512, MOSEI emotion at 512 / 768), likely due to over-parameterization and limited training data. In contrast, smaller dimensions (128) constrain the representation capacity and consistently underperform. An intermediate hidden size provides the optimal trade-off, avoiding underfitting at low dimensions and overfitting at high dimensions, thereby demonstrating the robustness of our framework across various datasets.

**Effect of Graph Depth and Attention Heads:** We study the impact of two key hyperparameters in our heterogeneous graph encoder: the number of layers ($L$) and the number of attention heads ($H$) (Table 10 Appendix).

*Graph Depth*: The shallow-to-moderate depth works best. IEMOCAP peaks at $L=3$ (73.26 WF1 / 73.25 ACC) for emotion, and 85.15 / 85.38 for sentiment, while MELD and MOSEI perform best with $L=2$. Deeper stacks ($L=4$) consistently overfit and degrade performance, while very shallow ($L=1$) models underfit.

*Attention Head*: ($H=4$) delivers the most reliable gains, recovering the best results across datasets. Too few heads ($H=1, 2$) limit the ability to capture diverse cross-modal relations, whereas too many ($H=8, 16$) introduce redundancy and reduce generalization. A moderate configuration ($L \in \{2, 3\}$ and $H=4$) is optimal, balancing representational capacity with robustness across benchmarks.

**Table 5:** Modality-level ablation for both **Emotion** (**E**) and **Sentiment** (**S**). Visual (**V**), Audio (**A**), and Text (**T**) are three modalities. Tri-modal fusion consistently outperforms uni- or bi-modal setups.

| V | A | T | IEMOCAP | | | | MELD | | | | CMU-MOSEI | | | |
|---|---|---|---|---|---|---|---|---|---|---|---|---|---|---|
| | | | E | | S | | E | | S | | E | | S | |
| | | | ACC | WF1 | ACC | WF1 | ACC | WF1 | ACC | WF1 | ACC | WF1 | ACC | WF1 |
| ✓ | ✗ | ✗ | 34.38 | 31.87 | 60.01 | 58.81 | 19.11 | 18.69 | 41.26 | 41.29 | 22.67 | 23.18 | 42.63 | 42.56 |
| ✗ | ✓ | ✗ | 54.59 | 52.18 | 59.64 | 58.95 | 34.90 | 35.19 | 48.54 | 46.56 | 42.34 | 42.12 | 60.76 | 60.59 |
| ✗ | ✗ | ✓ | 64.38 | 63.57 | 80.59 | 80.96 | 63.33 | 63.87 | 72.38 | 72.39 | 41.56 | 41.61 | 60.08 | 59.93 |
| ✓ | ✓ | ✗ | 57.54 | 56.73 | 70.73 | 71.55 | 62.64 | 64.11 | 73.49 | 73.43 | 28.57 | 28.39 | 47.95 | 47.90 |
| ✓ | ✗ | ✓ | 64.26 | 63.61 | 80.71 | 80.94 | 62.49 | 63.93 | 73.45 | 73.40 | 44.67 | 44.68 | 65.95 | 65.91 |
| ✗ | ✓ | ✓ | 67.52 | 67.74 | 80.03 | 80.92 | 62.80 | 64.23 | 73.10 | 73.09 | 43.43 | 43.68 | 65.45 | 65.67 |
| ✓ | ✓ | ✓ | **73.26** | **73.25** | **85.15** | **85.38** | **64.71** | **65.03** | **73.71** | **73.69** | **46.96** | **44.33** | **66.84** | **65.92** |

**Impact of Temporal Topology:** We evaluate our speaker-aware topology against standard $k$-step temporal neighborhoods (Table 6). On IEMOCAP and MELD, which provide speaker annotations, the speaker-aware design yields the best results (e.g., 73.26 ACC / 73.25 WF1 on IEMOCAP emotion; and 85.15 / 85.38 for MELD sentiment), consistently outperforming all $k$-step settings. By contrast, the $k$-step variant captures only temporal proximity and fails to model speaker-specific dependencies, for example, IEMOCAP sentiment drops to 81.95 / 82.58 at $k = 4$, well below the speaker-aware counterpart (85.15 / 85.38). For CMU-MOSEI, where speaker identity is unavailable, we rely on $k$-step neighborhoods and find that a moderate setting ($k = 4$) achieves the best trade-off (46.96 / 44.33 for emotion; 66.84 / 65.92 for sentiment). Smaller windows ($k = 2$) underutilize context, while larger ones ($K = 6$) admit noisy dependencies and degrade accuracy. This suggests speaker-aware topology is critical for modeling multi-party interactions, while in speaker-agnostic settings, a carefully tuned temporal neighborhood provides a strong alternative.

**Table 6:** $k$-step ablation on IEMOCAP, MELD, and CMU-MOSEI. For each dataset, we report ACC (left) and WF1 (right) for Emotion (E) and Sentiment (S).

| Step | IEMOCAP | | | | MELD | | | | CMU-MOSEI | | | |
|---|---|---|---|---|---|---|---|---|---|---|---|---|
| | Emotion | | Sentiment | | Emotion | | Sentiment | | Emotion | | Sentiment | |
| | ACC | WF1 | ACC | WF1 | ACC | WF1 | ACC | WF1 | ACC | WF1 | ACC | WF1 |
| $k=2$ | 69.01 | 69.03 | 80.71 | 81.62 | 63.68 | 63.88 | 71.03 | 71.00 | 45.89 | 45.55 | 66.28 | 65.12 |
| $k=4$ | 69.44 | 69.59 | 81.95 | 82.58 | 63.60 | 63.87 | 71.03 | 71.00 | **46.96** | **44.33** | **66.84** | **65.92** |
| $k=6$ | 69.75 | 69.95 | 81.89 | 82.54 | 63.56 | 63.85 | 70.84 | 70.83 | 46.14 | 44.18 | 65.89 | 65.30 |
| **Speaker-aware** | **73.26** | **73.25** | **85.15** | **85.38** | **64.71** | **65.03** | **73.71** | **73.69** | – | – | – | – |

**Impact of Unimodal Contextualization:** We evaluate the contribution of the unimodal transformer encoder (UniTrans), which contextualizes each modality independently before the heterogeneous graph construction (Table 7). Adding UniTrans yields substantial gains across all datasets: 73.26 / 73.25 (emotion) and 85.15 / 85.38 (sentiment) on IEMOCAP, 65.03 / 64.71 and 73.69 / 73.71 on MELD, and 44.33 / 46.96 and 66.84 / 65.92 on MOSEI. Removing UniTrans leads to marked degradation, most dramatically on MOSEI sentiment (52.39 / 52.69 vs. 66.84 / 65.92 with UniTrans). This suggests that contextualizing each modality with dedicated transformers is essential for capturing intra-modal dependencies, producing stronger unimodal embeddings that substantially enhance downstream cross-modal reasoning in the heterogeneous graph.

**Table 7:** Impact of a unimodal transformer (UniTrans) on Emotion (E) and Sentiment (S) across three datasets.

| Config | IEMOCAP | | | | MELD | | | | CMU-MOSEI | | | |
|---|---|---|---|---|---|---|---|---|---|---|---|---|
| | Emotion | | Sentiment | | Emotion | | Sentiment | | Emotion | | Sentiment | |
| | ACC | WF1 | ACC | WF1 | ACC | WF1 | ACC | WF1 | ACC | WF1 | ACC | WF1 |
| **w/ UniTrans** | **73.25** | **73.26** | **85.38** | **85.15** | **64.71** | **65.03** | **73.71** | **73.69** | **46.96** | **44.33** | **66.84** | **65.92** |
| **w/o UniTrans** | 68.70 | 68.56 | 83.12 | 83.53 | 61.95 | 63.16 | 71.76 | 71.95 | 33.63 | 30.55 | 52.39 | 52.69 |

## 5 CONCLUSION

We introduced a context-aware heterogeneous graph framework for multimodal emotion and sentiment analysis that learns relation-conditioned multimodal representations for emotion recognition. Unlike prior fusion models that combine modalities or rely on homogeneous message passing, our approach explicitly encodes structural heterogeneity (temporal, speaker-aware, and cross-modal edges) and semantic heterogeneity (relation- and type-specific transformations). Coupled with modality-specific Transformer encoders, this design enables fine-grained representation learning that generalizes across dyadic, multi-party, and monologue dialogue regimes. Experiments on IEMOCAP, MELD, and CMU-MOSEI show that our model achieves state-of-the-art results on IEMOCAP and MOSEI and remains competitive on MELD. It consistently outperforms recent Transformer and dialogue-specific approaches and slightly surpasses current graph-based methods. Ablations further confirm the value of heterogeneous relationship modeling, moderate hidden dimensionality, and balanced architectural depth.

Looking ahead, our findings suggest that explicitly modeling conversational structure and relational diversity is key to robust multimodal representation learning. Future work will extend this framework to richer multimodal streams (e.g., physiological or contextual signals), explore adaptive graph construction for real-time deployment, and investigate transfer to broader affective computing tasks.

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

APPENDIX

This appendix presents extended results, tables, and analyses supporting the main paper, including experimental details, hyperparameter studies, and additional t-SNE visualizations. We also disclose the use of LLMs as an assisting tool.

LLM USE

A large language model (LLM) was used solely as a support tool for minor copyediting (grammar, spelling, phrasing) to improve readability. It was not involved in generating research ideas, designing methods, running experiments, analyzing results, or substantive writing, and therefore should not be considered a contributor or author. All scientific content was written by the authors, and any LLM-suggested edits were incorporated only after human verification.

EMOTION-TO-SENTIMENT MAPPING.

We follow the existing methods such as GraphSmile Li et al. (2024b) and MMGCN Hu et al. (2021b), Our experiments use three sentiment categories (Negative, Neutral, Positive) obtained by merging dataset-specific emotion labels as summarized in Table 8: IEMOCAP maps {Sad, Angry, Frustrated}→Negative, Neutral→Neutral, and {Happy, Excited}→Positive; MELD uses its native Negative/Neutral/Positive; CMU-MOSEI maps {Highly Negative, Negative, Weakly Negative}→Negative, Neutral→Neutral, and {Weakly Positive, Positive, Highly Positive}→Positive.

**Table 8:** Merging scheme of emotion labels into sentiment categories across datasets.

| Sentiment | IEMOCAP | MELD | CMU-MOSEI |
|---|---|---|---|
| Negative | Sad, Angry, Frustrated | Negative | Highly Negative, Negative, Weakly Negative |
| Neutral | Neutral | Neutral | Neutral |
| Positive | Happy, Excited | Positive | Weakly Positive, Positive, Highly Positive |

EFFECT OF FEATURE DIMENSIONALITY.

**Table 9:** Appendix: Impact of feature dimension on performance for both **Emotion** and **Sentiment** on three datasets.

| Dim. | IEMOCAP | | | | MELD | | | | CMU-MOSEI | | | |
|---|---|---|---|---|---|---|---|---|---|---|---|---|
| | Emotion | | Sentiment | | Emotion | | Sentiment | | Emotion | | Sentiment | |
| | ACC | WF1 | ACC | WF1 | ACC | WF1 | ACC | WF1 | ACC | WF1 | ACC | WF1 |
| 128 | 67.09 | 67.63 | 82.37 | 82.92 | 62.64 | 64.02 | 72.75 | 72.77 | 42.44 | 42.29 | 57.86 | 60.15 |
| 256 | **73.26** | **73.25** | **85.15** | **85.38** | **64.71** | **65.03** | **73.71** | **73.69** | 43.65 | 43.04 | 64.60 | 65.38 |
| 512 | 70.91 | 71.02 | 83.48 | 83.89 | 62.14 | 63.67 | 72.60 | 72.74 | 39.96 | 39.68 | 60.56 | 62.37 |
| 640 | 72.15 | 72.21 | 83.18 | 83.68 | 62.07 | 63.71 | 73.10 | 73.16 | **46.96** | **44.33** | **66.84** | **65.82** |
| 768 | 71.60 | 71.56 | 84.72 | 85.13 | 63.10 | 64.16 | 72.91 | 72.89 | 41.34 | 41.77 | 63.14 | 63.94 |

EFFECT OF GRAPH DEPTH AND ATTENTION HEADS.

T-SNE VISUALIZATIONS ACROSS DATASETS

To qualitatively assess how the heterogeneous graph transformation (HGT) reshapes the representation space, we visualize raw, pre-HGT, and post-HGT embeddings with t-SNE for all datasets. For each dataset we show fused embeddings (Fig. 3, 5, 7) and modality-specific embeddings for text, audio, and visual (Fig. 4, 6, 8).

**IEMOCAP.** In the fused *raw* space (Fig. 3a), emotion categories overlap substantially, with weak boundaries between classes such as *happy* and *neutral*. The *pre-HGT* projection (Fig. 3b) begins to disentangle regions and tightens clusters for distinct classes (e.g., *sad*, *angry*). The *post-HGT* embeddings (Fig. 3c) yield the clearest separation: clusters are compact with larger margins, indicating improved intra-class compactness and inter-class separation. Modality-wise (Fig. 4), raw

**Table 10:** Appendix Impact of #Layers and #Heads in our heterogeneous graph for both **Emotion** and **Sentiment**.

| | Config. | IEMOCAP | | | | MELD | | | | CMU-MOSEI | | | |
| | | Emotion | | Sentiment | | Emotion | | Sentiment | | Emotion | | Sentiment | |
| | | ACC | WF1 | ACC | WF1 | ACC | WF1 | ACC | WF1 | ACC | WF1 | ACC | WF1 |
|---|---|---|---|---|---|---|---|---|---|---|---|---|---|
| **Layers** | $L = 1$ | 70.43 | 70.68 | 84.47 | 84.90 | 62.95 | 63.87 | 72.91 | 73.00 | 42.61 | 42.70 | 63.93 | 64.06 |
| | $L = 2$ | 71.72 | 71.75 | 84.84 | 85.13 | **65.03** | **64.71** | **73.69** | **73.71** | **46.96** | **44.33** | **65.82** | **66.84** |
| | $L = 3$ | **73.25** | **73.26** | **85.38** | 85.15 | 62.22 | 63.74 | 72.11 | 72.29 | 42.35 | 42.19 | 61.18 | 62.81 |
| | $L = 4$ | 68.70 | 69.13 | 82.19 | 82.80 | 62.60 | 64.17 | 72.91 | 73.00 | 42.32 | 41.68 | 58.29 | 59.73 |
| **Heads** | $H = 1$ | 71.04 | 71.27 | 83.24 | 83.69 | 63.26 | 64.29 | 72.76 | 72.83 | 41.73 | 40.84 | 54.46 | 56.97 |
| | $H = 2$ | 71.29 | 71.56 | 85.40 | 85.70 | 62.10 | 63.59 | 72.26 | 72.37 | 40.17 | 40.79 | 61.47 | 62.98 |
| | $H = 4$ | **73.26** | **73.25** | **85.15** | **85.38** | **65.03** | **64.71** | **73.69** | **73.71** | **46.96** | **44.33** | **65.82** | **66.84** |
| | $H = 8$ | 70.12 | 70.29 | 82.44 | 83.00 | 62.79 | 64.31 | 73.06 | 72.94 | 43.11 | 43.08 | 62.69 | 64.16 |
| | $H = 16$ | 70.43 | 70.61 | 82.99 | 83.63 | 62.75 | 64.19 | 72.41 | 72.52 | 44.09 | 42.38 | 59.91 | 61.89 |

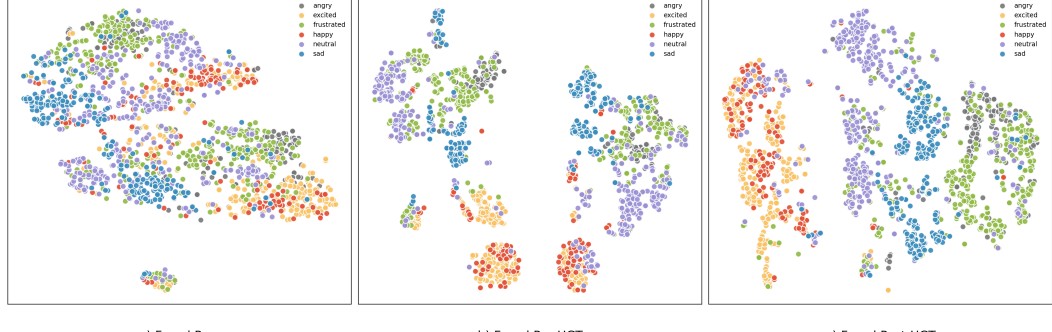

a) Fused Raw        b) Fused Pre-HGT        c) Fused Post-HGT

**Figure 3:** IEMOCAP fused embeddings with t-SNE: (a) Raw, (b) Pre-HGT, (c) Post-HGT.

audio and visual features are highly mixed, whereas text shows partial separability. Pre-HGT consistently improves all three modalities, and post-HGT further sharpens boundaries—most prominently for text—and also reduces confusion between hard pairs such as *frustrated* vs. *neutral* in audio.

**CMU-MOSEI.** The fused *raw* embeddings (Fig. 5a) are heavily entangled, especially for mid-range sentiments (*slightly positive/negative*) that blur with *neutral*. *Pre-HGT* (Fig. 5b) tightens structure for the extremes (*very positive*, *very negative*) but overlaps persist elsewhere. *Post-HGT* (Fig. 5c) produces the most discriminative space: fine-grained distinctions between *slightly positive*, *positive*, and *very positive* become visible and *neutral* occupies a more coherent region. In the unimodal plots (Fig. 6), text emerges as the strongest signal after HGT with well-defined clusters; audio benefits from HGT mainly at the extremes; visual improves but remains less separable than text and audio. The fused post-HGT space clearly outperforms any single modality.

**MELD.** In the fused *raw* space (Fig. 7a), conversational noise leads to broad overlaps among *neutral*, *anger*, and *sadness*. *Pre-HGT* (Fig. 7b) increases local coherence; *joy* and *surprise* begin to separate while *neutral* remains diffuse. *Post-HGT* (Fig. 7c) yields the clearest structure with dense clusters for *anger*, *sadness*, and *joy*, and a more stable region for *neutral*. At the modality level (Fig. 8), text provides the most discriminative embeddings from raw to post-HGT. Audio starts highly entangled, gains some structure pre-HGT, and shows clearer extreme-class regions post-HGT, though fine-grained distinctions remain challenging. Visual follows a similar trend—improved organization post-HGT yet still noisier than text. Their complementary cues, however, are effectively leveraged in the fused post-HGT space.

Across all datasets, the trajectory *raw → pre-HGT → post-HGT* consistently transforms overlapping clouds into compact, well-separated clusters. Text is the most discriminative unimodal source, while audio and visual are noisier yet complementary. The fused post-HGT embeddings achieve the strongest separation, corroborating the quantitative gains by showing that HGT captures cross-modal interactions to build richer and more class-discriminative representations.

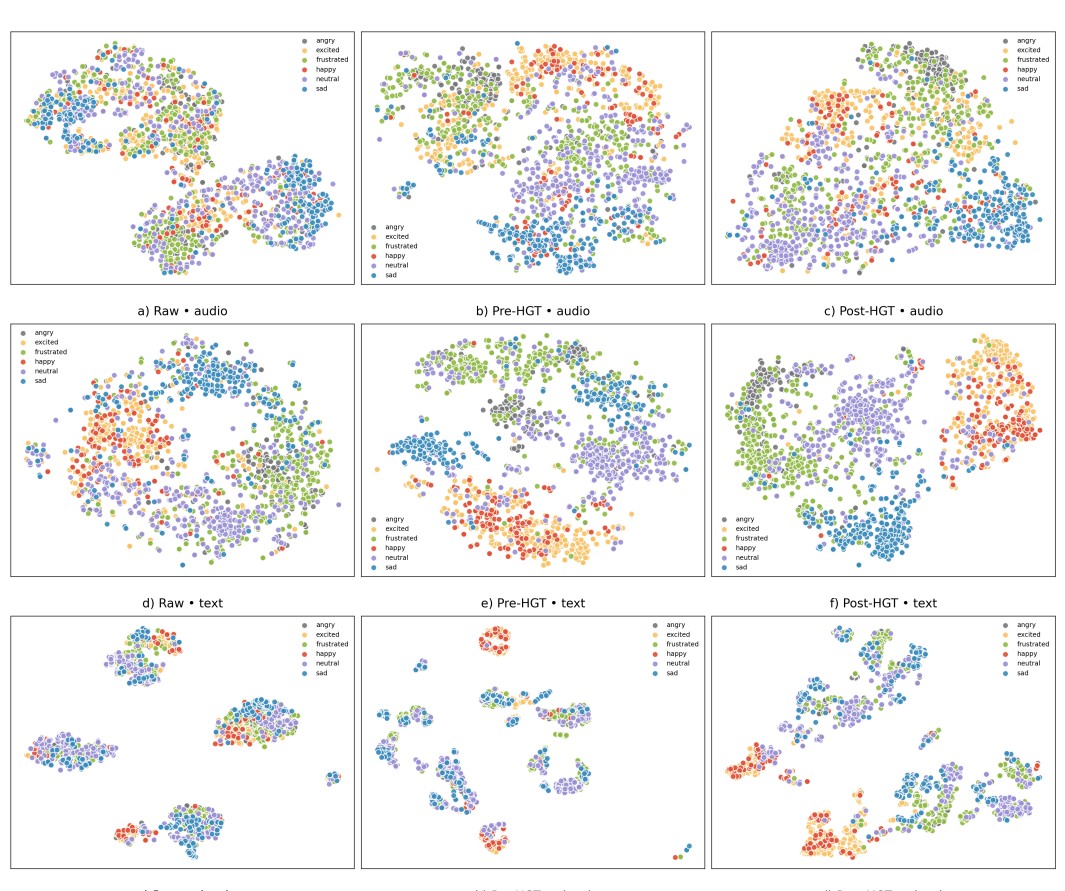

**Figure 4:** IEMOCAP modality-specific embeddings (text, audio, visual) across stages: raw, pre-HGT, and post-HGT.

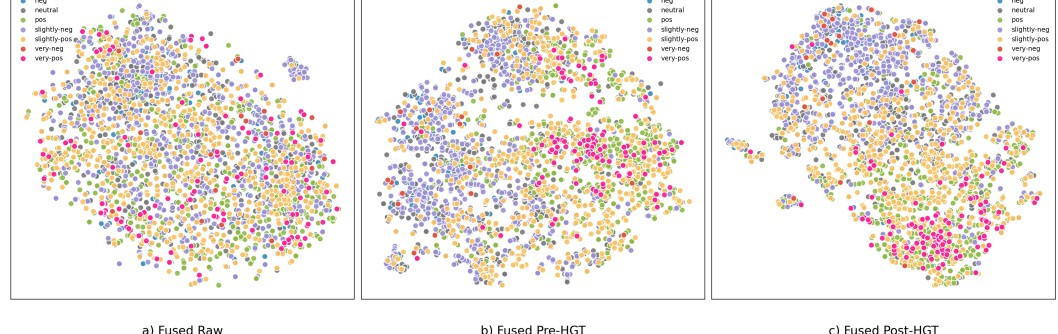

**Figure 5:** CMU-MOSEI fused embeddings with t-SNE: (a) Raw, (b) Pre-HGT, (c) Post-HGT.

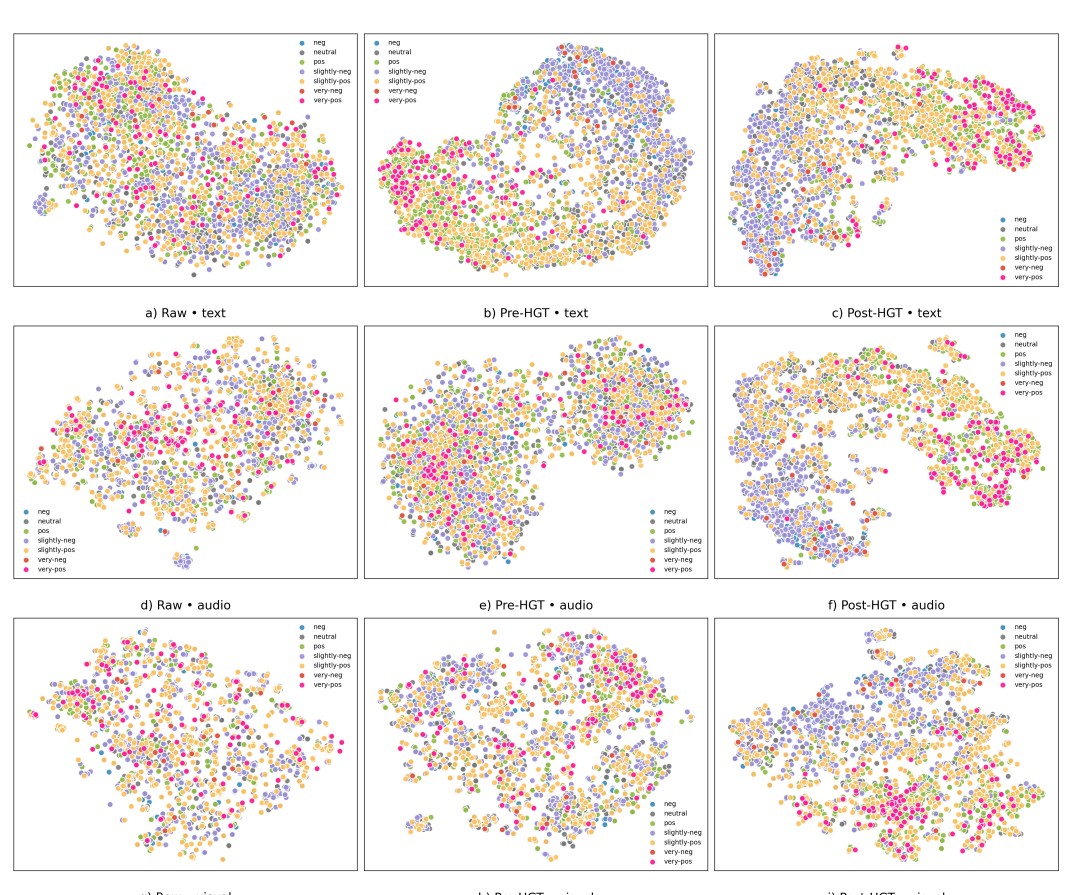

**Figure 6:** CMU-MOSEI modality-specific embeddings (text, audio, visual) across stages: raw, pre-HGT, and post-HGT.

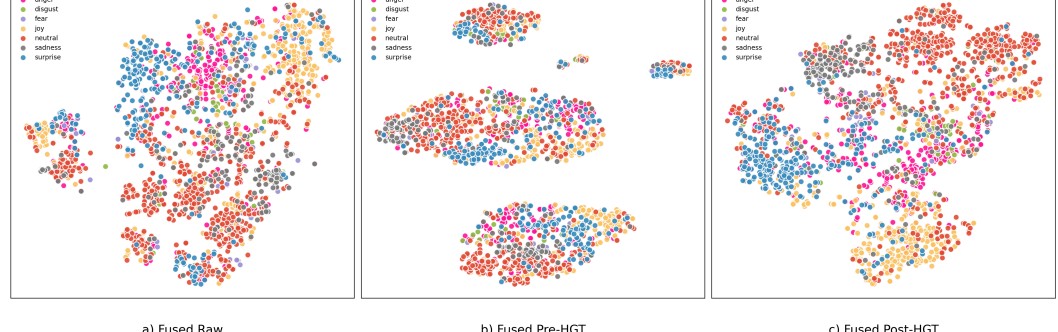

**Figure 7:** MELD fused embeddings with t-SNE: (a) Raw, (b) Pre-HGT, (c) Post-HGT.

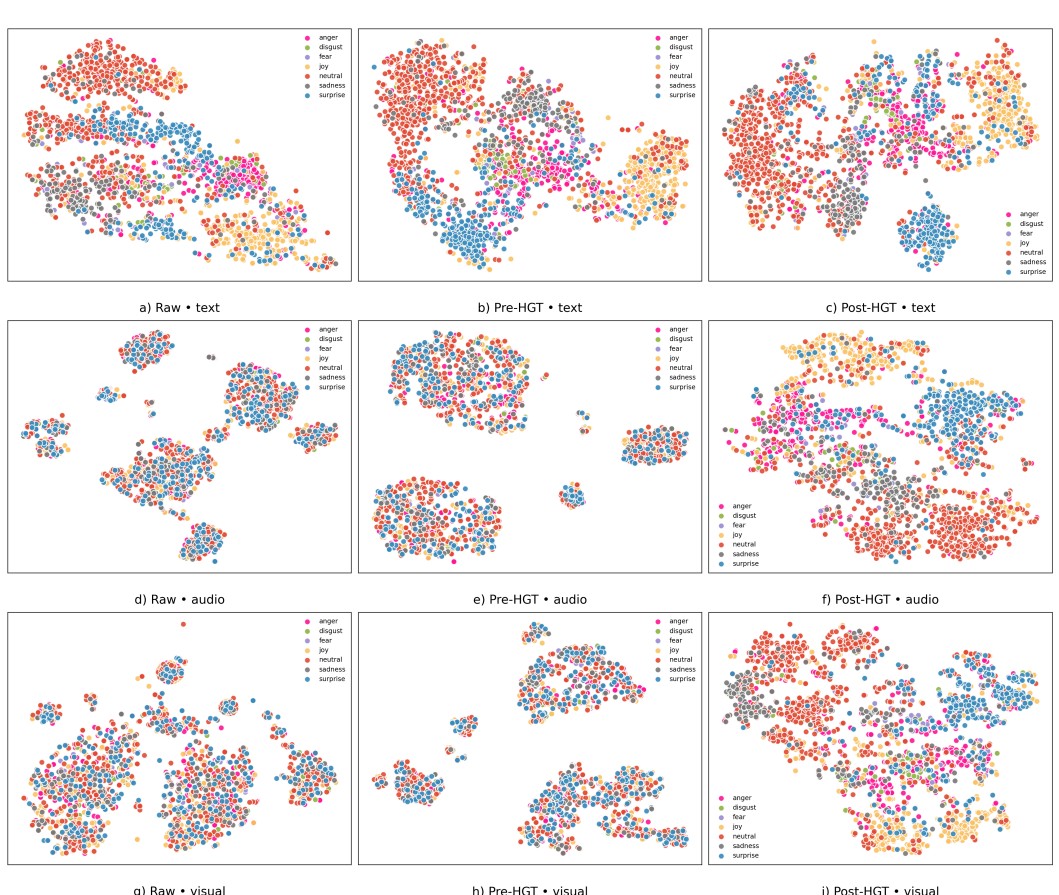

**Figure 8:** MELD modality-specific embeddings (text, audio, visual) across stages: raw, pre-HGT, and post-HGT.

