# OpenReview forum: "Context-aware Heterogeneous Graph-driven Multimodal Representation Learning for Emotion Recognition"
_ICLR.cc/2026/Conference — Submitted to ICLR 2026_

### Official Review · Reviewer_rnw7 · 2025-10-24

**Soundness:** 3
**Presentation:** 2
**Contribution:** 2
**Rating:** 2
**Confidence:** 4

**Summary:**

The paper proposes a heterogeneous graph-based multimodal emotion recognition model that connects textual, acoustic, and visual features through relation-aware attention across temporal and cross-modal links. It effectively supports both single- and multi-speaker dialogues and achieves state-of-the-art results on the IEMOCAP and CMU-MOSEI datasets.

**Strengths:**

1. The motivation is sound, as the model incorporates typed temporal, cross-modal, and relation-conditioned projections shared across layers. This design goes beyond conventional homogeneous GAT/GNN fusion and standard sequence attention mechanisms.
2. The framework is versatile, effectively supporting both multi-speaker and single-speaker scenarios.
3. The proposed method achieves state-of-the-art performance on the IEMOCAP and CMU-MOSEI datasets.

**Weaknesses:**

1. This work differs from previous studies in that it treats each modality within every utterance as individual graph nodes for attention learning. However, this design naturally leads to excessive information aggregation, which may dilute the model’s focus on important nodes or modalities. Moreover, it can cause the model to become overly dependent on these aggregated features. In real-world scenarios, modalities are often missing or ambiguous, which could easily lead to model failure.
2. The main innovation lies in the design of the heterogeneous graph and the relation-aware attention mechanism. The former simply integrates multiple modality nodes into a unified graph, while the latter modifies the standard GAT input from homogeneous to heterogeneous. These adjustments cannot be considered major innovations. I would expect to see more originality in the graph structure itself, rather than merely combining all data into a single graph and allowing the model to learn attention automatically. Furthermore, the performance improvement is marginal, 0.4 on IEMOCAP and 0.6 on CMU-MOSEI, with no SOTA on MELD, and the absence of statistical analysis makes it difficult to justify the claimed contributions.
3. Figure 1 and Figure 2 essentially illustrate the same concept, and there is no need to include both. The authors should focus more on explaining the proposed method in depth. For instance, the most critical component, the Heterogeneous Graph Encoder, is only described in half a page. Additionally, the authors should elaborate on the originality and underlying principles of the proposed graph structure. In MERC, the receptive field of the graph is crucial; when handling long conversations, the model should pay more attention to nearby utterances, but the paper fails to explain how this is achieved.

**Questions:**

1. Provide a more detailed explanation of the advantages of the proposed graph structure, including why it outperforms existing graph designs and why it is necessary. For example, it would be helpful to illustrate how the model differentiates its attention toward heterogeneous information and how it effectively mitigates issues of information redundancy and dilution.
2. Explain how the model handles long-term conversational dependencies and assess its robustness to missing modalities. Such clarification would enhance the understanding of the model’s applicability to real-world, imperfect multimodal data.
3. The authors claim that the proposed model offers interpretability. It would significantly strengthen the paper if they could provide concrete examples, such as how the model identifies and prioritizes key information within a dialogue, to illustrate the interpretive capacity and validate the effectiveness of the proposed approach.

---

> ### Author Response · Authors · 2025-12-04
> **Acknowledgment of Review and Future Plans**
>
> We sincerely thank Reviewer rnw7 for their thoughtful assessment of Submission 21069. We appreciate your recognition of the framework's versatility in supporting both single- and multi-speaker scenarios and its performance on IEMOCAP and CMU-MOSEI.
>
> We have carefully considered your concerns regarding the potential for excessive information aggregation and the risk of the model becoming overly dependent on aggregated features. We also acknowledge the need to provide a more rigorous statistical analysis to justify the performance improvements and to elaborate further on the Heterogeneous Graph Encoder. We plan to address the redundancy between Figures 1 and 2 and expand the explanation of the graph structure as suggested.
>
> We will utilize these insights to refine the paper significantly and submit the improved version to an upcoming conference.

---

### Official Review · Reviewer_yZQF · 2025-10-28

**Soundness:** 2
**Presentation:** 2
**Contribution:** 2
**Rating:** 2
**Confidence:** 4

**Summary:**

This paper argues that existing methods overlook modality-specific temporal dynamics and asymmetric dependencies. To address these issues, the authors propose a heterogeneous GNN that encodes both structural and semantic heterogeneity. Experiments on IEMOCAP, MELD, and CMU-MOSEI demonstrate state-of-the-art performance.

**Strengths:**

1. The topic of the paper is interesting, and the authors correctly identify that existing methods fail to consider all relevant factors within a unified framework. We find the motivation well-grounded.

2. The analysis of the experimental results is concise and clearly presented.

3. The conclusion provides insightful directions for future work.

**Weaknesses:**

1. From the Introduction section, the research motivation of this paper appears to be incremental — aiming to jointly consider both structural and semantic heterogeneity. However, the paper mainly redefines the types and updating mechanisms of edges within a heterogeneous network, without departing from the conventional heterogeneous-graph-based framework for multimodal emotion recognition. Therefore, we believe the novelty of this work is limited.

2. The Methodology section is unclear and lacks a detailed description of the training loss, which we consider an essential part of the method.

3. The paper is missing crucial ablation studies. It does not provide evidence showing whether the model benefits from the proposed structural heterogeneity or from the semantic heterogeneity. In particular, an ablation analysis on the edge construction in Figure 2 is absent.

4. The paper lacks a comparison and analysis of time complexity. Since the authors employ both Transformer and GNN components, we suspect that the proposed approach may not scale well to large models. Including comparisons of runtime or efficiency with existing methods would significantly strengthen the work.

5. The reported results are not as strong as those of GraphSmile, and on the IEMOCAP dataset, some results even fall behind the baselines. Thus, the authors’ claim of achieving state-of-the-art performance seems overstated.

**Questions:**

Please refer to the above-mentioned weaknesses.

---

> ### Author Response · Authors · 2025-12-04
> **Acknowledgment of Review and Future Plans**
>
> We sincerely thank Reviewer yZQF for their assessment of Submission 21069. We appreciate your recognition that our research motivation is well-grounded and that the experimental analysis is presented clearly.
>
> We have carefully noted your concerns regarding the perceived incremental nature of the work and the clarity of the Methodology section. We acknowledge that the description of the training loss and the time complexity analysis are missing and are essential for a complete evaluation. We plan to add these details and the suggested ablation studies on edge construction to the next version of the paper.
>
> We will use this feedback to significantly refine the manuscript and submit the improved work to an upcoming conference.

---

### Official Review · Reviewer_ch2m · 2025-10-28

**Soundness:** 2
**Presentation:** 2
**Contribution:** 2
**Rating:** 2
**Confidence:** 4

**Summary:**

This paper proposes a context-aware heterogeneous graph framework for multimodal emotion recognition (MER).
The authors argue that existing models fail to distinguish between different types of interactions
The proposed method first uses standard Transformer encoders to contextualize each modality independently.
Then, it constructs a heterogeneous graph where nodes represent modal features and edges are typed to distinguish between temporal, cross-modal, and speaker-conditioned links .
A relation-aware graph transformer performs message passing using different transformations for each edge type.
The model adapts its topology for multi-party dialogue versus single-speaker monologues.

**Strengths:**

1. Transformers and homogeneous GNNs treat all interactions uniformly. Explicitly modeling different relationship types as a heterogeneous graph is a well-motivated approach

2. The model's design cleverly adapts to the input data. It uses speaker-aware edges for multi-party dialogues and k-step temporal edges for monologues, showing flexibility.

3. The paper provides useful ablation studies.

**Weaknesses:**

1. The core relation-aware graph transformer is close to a standard implementation of a Heterogeneous Graph Transformer, reduce the novelty.

2. The paper notes it is competitive but below GraphSmile for emotion (Table 2) and slightly below DialogueCRN for sentiment (Table 4).     This undermines the claim of superiority and suggests the model may not fully handle the multi-party complexity.

3. The model is a two-stage pipeline: a stack of Transformer encoders (one for each modality) followed by a stack of heterogeneous graph layers. This serial design is parameter-heavy and computationally complex compared to more integrated approaches.

4. The ablation study shows that removing the unimodal Transformer pre-processing causes a catastrophic performance drop.  For example, on CMU-MOSEI sentiment, accuracy falls from 66.84 to 52.39. This suggests that the initial Transformer encoders are responsible for the vast majority of the performance, and the heterogeneous graph component may only provide a marginal improvement, reducing its effectiveness.

5. The t-SNE visualizations do not convincingly show a major improvement from the graph.

6. This late-fusion step is simple and may be a bottleneck, as it does not allow for higher-order interactions between the contextualized modal embeddings before classification.

**Questions:**

See weakness.

---

> ### Author Response · Authors · 2025-12-04
> **Acknowledgment of Review and Future Plans**
>
> We sincerely thank Reviewer ch2m for their detailed assessment of Submission 21069. We appreciate your positive comments regarding the motivation behind modeling relationship types as a heterogeneous graph and the flexibility of our topology design.
>
> We have carefully considered your feedback regarding the novelty of the relation-aware graph transformer and the model's computational complexity. We specifically acknowledge your concern that the initial Transformer encoders may be responsible for the majority of the performance, as highlighted by the ablation studies. We plan to address these architectural bottlenecks and validate the graph component's contribution more rigorously.
>
> We will incorporate these improvements to strengthen the paper and submit the revised work to an upcoming conference.

---

### Official Review · Reviewer_rNLE · 2025-10-31

**Soundness:** 3
**Presentation:** 3
**Contribution:** 2
**Rating:** 2
**Confidence:** 4

**Summary:**

The paper addresses multimodal emotion recognition (MER) challenges where existing models ignore modality-specific dynamics and asymmetric dependencies. It proposes a context-aware heterogeneous graph-driven method: using dedicated Transformer encoders for unimodal features, a relation-aware graph transformer for message passing, and adaptive topology for different datasets. Experiments show it outperforms state-of-the-art, verifying heterogeneity’s necessity.

**Strengths:**

The results of the experiment are promising.

**Weaknesses:**

1. The proposed relation-aware attention mechanism differs mathematically and conceptually from previous works such as HMG-Emo or HHGN.
2. Are there theoretical guarantees or formal proofs supporting the claim that the model better captures structural and semantic heterogeneity?
3. While the paper consistently outperforms transformer-based (e.g., Joyful, SACL-LSTM) and dialogue-specific baselines (e.g., DialogueCRN, MMGCN), the margin of improvement over graph-based methods like GraphSmile and HHGN is relatively small in certain benchmarks (e.g., MELD).
4. Extending the experimental evaluation to other multimodal sentiment datasets (such as EmoReact or MuSE) or missing modality scenarios will further validate the generalization ability of the proposed method.

**Questions:**

See Weaknesses.

---

> ### Author Response · Authors · 2025-12-04
> **Acknowledgment of Review and Future Plans**
>
> We sincerely thank Reviewer rNLE for their time and constructive feedback on Submission 21069. We value your assessment of our work on multimodal emotion recognition.
>
> We explicitly acknowledge the weaknesses highlighted, particularly regarding the need for formal theoretical guarantees and the comparison with graph-based baselines. We plan to carefully incorporate your suggestions—specifically clarifying the mathematical distinctions of our attention mechanism and extending our evaluation to additional datasets like EmoReact or MuSE.
>
> We will utilize these insights to refine the paper significantly and submit the improved version to an upcoming conference. Thank you again for helping us strengthen our research.

---

### Meta-Review · Area_Chair_YvEg · 2026-01-08

**Summary:**

This paper proposes a heterogeneous graph model for multimodal emotion recognition. While the approach is motivated and shows competitive results, all reviewers raise substantial concerns. Key issues include limited novelty (the core graph transformer resembles existing designs), insufficient evidence of the graph component’s contribution beyond the initial Transformer encoders, marginal gains over strong graph baselines, and missing analyses (theoretical guarantees, time complexity, robustness to missing modalities). The authors acknowledge the concerns but do not provide new experiments or clarifications in the rebuttal. Thus, the work in its current form does not convincingly advance the state of the art or adequately address its methodological claims.

**Reviewer Concerns:**

The authors’ replies acknowledge all concerns but do not substantively address them. Outstanding issues include: (1) limited novelty of the relation-aware graph transformer; (2) unclear contribution of the graph component vs. the heavy unimodal Transformers; (3) marginal improvements over graph baselines; (4) missing ablation studies on edge construction, time complexity, and theoretical grounding; (5) insufficient discussion of robustness and interpretability. The rebuttal offers only plans for future work, leaving the core criticisms unresolved.

**Reviewer Scores:**

Given the authors’ replies did not include new evidence or analyses to mitigate the concerns, it is unlikely any reviewer would raise their score. All reviewers (rNLE, ch2m, yZQF, rnw7) expressed consistent criticisms with high confidence. Their primary issues—novelty, ablation support, and empirical rigor—remain unaddressed. Thus, their original ratings (2: reject) would likely stay unchanged after discussion.

---

### Decision · Program_Chairs · 2026-01-26

Reject